# Effects of Manganese and Zirconium Dispersoids on Strain Localization in Aluminum Alloys

Elena Jover Carrasco [1], Juliette Chevy [2], Belen Davo [2] and Marc Fivel [1,*]

[1] SIMaP, University Grenoble Alpes, CNRS, 38000 Grenoble, France; Elena.Jovercarrasco@grenoble-inp.fr
[2] C-TEC Constellium Technology Center, Metallurgy Department, 38341 Voreppe CEDEX, France; Juliette.Chevy@constellium.com (J.C.); Belen.Davo@constellium.com (B.D.)
* Correspondence: Marc.Fivel@grenoble-inp.fr; Tel.: +33-0476826336

**Abstract:** Strain localization in aluminum alloys can cause early failure of the material. Manganese and zirconium dispersoids, often present in aluminum alloys to control the grain size, have been found to be able to homogenize strain. To understand the effects of dispersoids on strain localization, a study of slip bands formed during tensile tests is carried out both experimentally and through simulations using interferometry and discrete dislocations dynamics. Simulations with various dispersoid size, volume fraction, and nature were carried out. The presence of dispersoids is proven to homogenize strain both is the experimental and numerical results.

**Keywords:** plasticity; slip bands; discrete dislocations dynamics; Al-Cu-Li alloys; Mn dispersoids; Zr dispersoids





## 1. Introduction

The aerospace industry has largely benefited from the development of optimized aluminum alloys, which could maximize the performances of structural parts for minimum weight. Recently developed Al-Cu-Li alloys optimally meet these challenging aeronautic specifications requiring both improved strength and damage tolerance [1,2].

One important factor influencing damage tolerance is strain localization. In the case of cyclic loading, it has been shown that planar slip can be advantageous to prevent fatigue crack propagation, since it leads to more reversibility in the accumulated plastic strain [3]. On the other hand, in the case of monotonic loading, strain localization should be avoided, since the stress concentration induced at the grain boundaries could lead to premature failure. Strain localization is influenced by different microstructural parameters, such as hardening precipitation, solid solutions, grain size, and orientation (texture).

In Aluminum alloys, Li addition leads to the formation of the strengthening $\delta'$-$Al_3Li$ precipitates, while the addition of Cu together with Li leads to the additional formation of the more stable $T_1$-$Al_2CuLi$ phase. Both phases are reported to be shear-able, but the shearing mechanism differs—leading to significant differences in the strain localization process [4]. These localization mechanisms and the resulting properties regarding of fatigue life and toughness have been widely studied [5,6].

Certain addition elements like Zr, Mn, Sc, or Cr are used to control the grain morphology. They precipitate into dispersoids, which consist of fine intermetallic particles [7]. They typically form during the homogenization process, and their size varies from 10 nm to 200 nm depending on their nature and the homogenization parameters [8]. The main purpose of the dispersoids is to pin the grain boundaries to control grain size and recrystallization of the material during the various stages of the alloy fabrication and to form, but they also play a major role in toughness and fatigue resistance as they influence strain localization through their interactions with dislocations [1,2,8,9]. This paper investigates the effect of these dispersoids on strain localization. In this work, we will focus on Mn and Zr dispersoids as they are the most commonly found in commercial Al-Cu-Li alloys.

Manganese dispersoids are present in aluminum alloys in the form of $Al_{20}Cu_2Mn_3$ ellipsoidal particles that can be up to 500 nm long by 100 nm large. Manganese being a eutectic, manganese dispersoids are concentrated at the grain boundaries [8]. In rolled samples, manganese dispersoids are mostly aligned and can act as nucleation sites for precipitates [10]. Manganese dispersoids are reported to spread and homogenize the plastic strain in aluminum alloys, increasing the toughness of said alloy [11]. Manganese dispersoids are incoherent with the matrix, which can lead to microvoid coalescence reducing the fracture toughness [11–13], and thus, counterbalancing the positive effect of manganese on the mechanical properties of the alloys.

Zr-containing dispersoids, in the from $Al_3Zr$, vary in number density within a given aluminum alloy, creating precipitate rich and precipitate free zones (PFZ) around the cast grain boundaries, due to zirconium's peritectic behavior in aluminum alloys. These dispersoids are usually sphere-shaped with a radius of about 20 nm [14]. Due to their small size, when the applied strain is increased, they are first by-passed by the dislocations forming Orowan loops around them, and they are later sheared when the applied strain/stress becomes high enough [15]. Zirconium effect on strain localization is less known. Due to the small size of the zirconium dispersoids, their effect on strain homogenization is less prominent than the manganese dispersoids [16].

Strain localization is associated with the formation of slip bands inside the material. The slip bands are shown by the slip traces printed on the surface of the specimen [17,18]. They are formed by dislocation activity in close slip planes. Due to its high stacking fault energy ($150 \, \text{mJ} \cdot \text{m}^{-2}$ [19]), the thermally activated cross-slip mechanism is very frequent in Aluminum. This mechanism augments the number of slip systems available to achieve large strains, which increases the ductility. Obviously, the cross-slip probability strongly affects the mechanisms of slip band multiplication. As an example, double cross-slip will give the possibility of dislocations to by-pass particles. It will also grow the slip bands by forming Koehler sources.

As said before, the formation of slip bands is controlled by many parameters related to the alloy composition. Texture and loading direction are other influential parameters. For aluminum alloys, texture components Brass, Copper, R, and S form more slip bands than Goss, Q, or P [20]. It has also been shown that the orientation of the tensile axis has an effect on the evolution of the slip bands [21]. AFM observation of Al single crystal has demonstrated that [001] uniaxial loading leads to a saturation in the slip band activity after a few percent of plastic strain. Inversely, [112] loading direction leads to continuously increasing localization in the slip bands.

This paper aims to elucidate the individual contribution of Mn and Zr dispersoids on the strain localization in Al-Cu-Li alloys using a combination of experiments and simulations. The experiments consist of tensile tests performed on Al-Cu-Li alloys specially designed to contain the investigated Zr and Mn dispersoids. The simulations are performed using 3D Discrete Dislocations Dynamics (DDD). DDD is indeed the ideal tool to address the complex dislocation/particle interaction mechanism [22], since it intrinsically contains the physics of the cross-slip mechanism as opposed to Finite Element of full field Fast Fourier Transform modeling, which could only take care of cross-slip via ad-hoc equations in the constitutive model. As an example, DDD simulations have successfully shown the role of particles in strain localization in fatigue [23].

The effect of each type of dispersoids, their size, and volume fraction on the strain localization will be investigated and compared to experiments. Both for the simulations and the experiments, the slip localization is measured from line profiles.

## 2. Materials and Methods

### 2.1. Experimental Procedure

Four alloys of the Al-Cu-Li alloy family in T8 condition were used in this study. The strain localization of "high Li" extruded products and "Low Li" rolled products was examined containing either only Zr dispersoids, or both Zr and Mn dispersoids, as

indicated in Table 1. The microstructure was mainly unrecrystallized for all of them, and they presented typical deformation texture. Four samples were tested as described in the following table.

**Table 1.** Composition of the four samples investigated in the experiments.

| Name | Zr Dispersoids | Mn Dispersoids | wt% | | | |
|---|---|---|---|---|---|---|
| | | | Li | Cu | Mn | Zr |
| Low Li, Zr-Mn | Yes | Yes | 0.9 | 4.3 | 0.33 | 0.13 |
| Low Li, Zr | Yes | No | 0.9 | 4.3 | - | 0.14 |
| High Li, Zr-Mn | Yes | Yes | 1.6 | 3.0 | 0.31 | 0.14 |
| High Li, Zr | Yes | No | 1.6 | 3.0 | - | 0.14 |

In order to observe slip bands, sample surfaces were mirror-polished prior to their deformation to eliminate the sample's preexisting roughness and to ensure that the slip bands measured are independent of the surface state. First, 160 mm long aluminum samples were machined in a TOP 11 shape with a gauge length of 54 mm and a 12 mm × 2 mm section. The samples, coming from rolled products, were machined with the observation plane corresponding to the L-TL plane. This surface orientation was chosen due to the larger grain size in that particular plane, which visualizes longer slip bands.

To study the effect of dispersoids, other factors impacting the slip band formation need to be controlled. Textures found in rolled samples are limited, making it easier to find grains with similar orientations on different samples.

A tensile strain is then applied along the L direction of the samples at a displacement rate of 20 mm/min up to a total tensile strain of 4.3%. This strain was chosen to be high enough to induce visible slip bands at the free surface but low enough for the sample to not be overrun by slip bands.

Once the slip bands are visible, the samples are observed with differential interference contrast microscopy to qualitatively study the slip band organization and to select zones of interest to be further studied. As shown in Figure 1, slip bands are easily observed, and their direction and intensity depend on the grain orientation. These selected zones were first scanned in EBSD to determine the grain orientations. Once the orientation was known, those zones were mapped using an interferometer to quantify the roughness induced by the slip bands at the surface. Ten profiles perpendicular to the slip bands are then taken and analyzed in each grain.

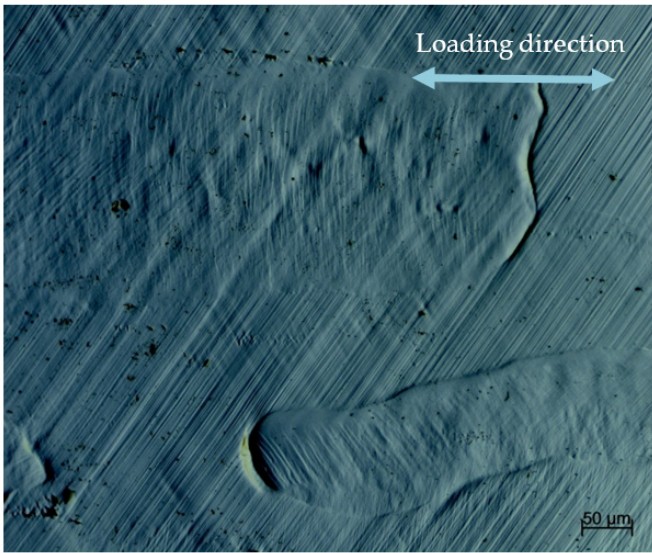

**Figure 1.** Typical slip bands formed in Aluminum alloy after 4.3% straining observed with differential interference contrast microscopy.

It is worth mentioning that the signal treatment of the interferometry results was kept to a minimum to ensure slip bands remained visible. Finally, the mean height of profile elements ($P_c$) is calculated together with the spacing between the bands $P_{sm}$ (See Figure 2). The magnitude of the strain localization is finally quantified by the local shear, $\gamma_{loc}$, defined as the ratio $P_c/P_{sm}$.

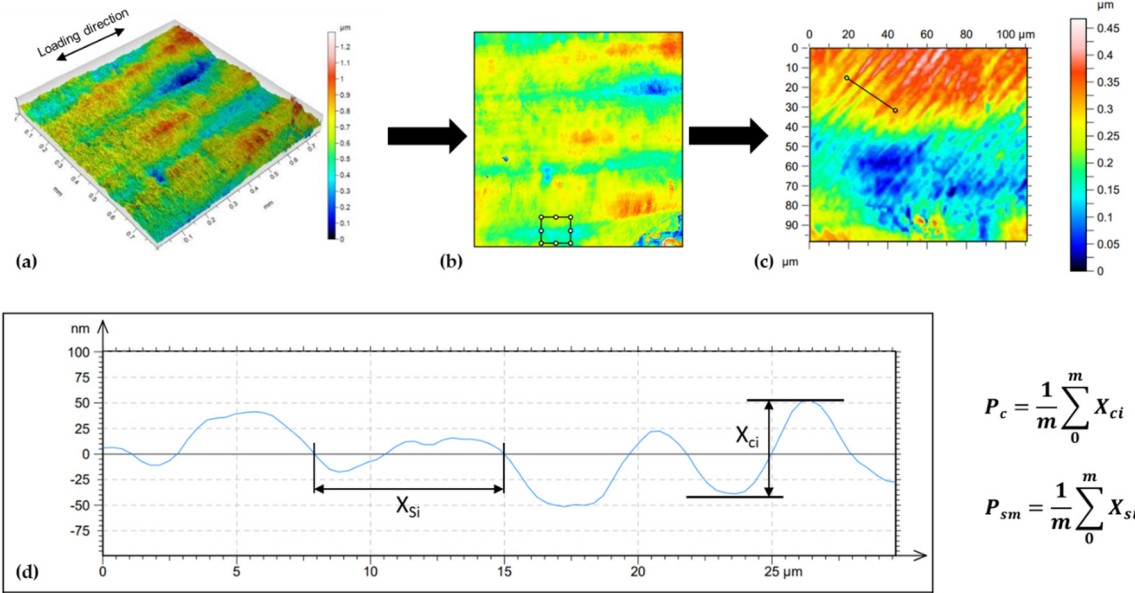

$$P_c = \frac{1}{m} \sum_0^m X_{ci}$$

$$P_{sm} = \frac{1}{m} \sum_0^m X_{si}$$

**Figure 2.** Interferometric mapping of a High-Li aluminum alloy. (**a**) 0.8 mm × 0.8 mm 3D surface mapping. (**b**) In plane projection of the surface heights. (**c**) zoom on a 100 μm × 100 μm area. (**d**) Typical line profile is taken perpendicular to the slip bands from which the quantities $P_c$ and $P_{sm}$ are evaluated.

### 2.2. Dislocation Dynamics Simulations

DDD simulations are performed using the edge-screw model TRIDIS [24] with the material parameters corresponding to Aluminum recalled in Table 2. Note that this code is precisely the one that was previously used to address dislocation-precipitate interactions in Reference [22], and applied to the case of fatigue of Waspaloys [23] and creep of Ni Superalloys [25].

**Table 2.** Discrete dislocations dynamics (DDD) simulation parameters.

| Burgers Vector | Shear Modulus | Time Step | Cross-Slip Parameters | | |
|---|---|---|---|---|---|
| b [Ang.] | G [GPa] | $\delta t$ [s] | $\tau_{\mathrm{III}}$ [MPa] | V [$b^3$] | T [K] |
| 2.86 | 32 | $10^{-10}$ | 32 | 350 | 300 |

The DDD principle is detailed in Reference [24], but the main ingredient used in TRIDIS are recalled hereafter. Dislocation lines are discretized in piece-wise edge and screw segments. At each time step, the effective resolved shear stress is evaluated at the segment centers as the superposition of the applied stress and the internal stress induced by all the segments contained in the simulated volume. In the present simulations, the applied stress tensor is taken as homogeneous and corresponds to a pure tensile load.

The enforced boundary conditions aim at representing an isolated grain embedded in a much bigger polycrystalline sample. The grain boundaries consist of sets of facets that prevent the dislocations from escaping the simulated box. Here the simulated volume consists of a spherical grain of 5 μm of diameter approximated by a set of 24 facets. The particles are also geometrically defined by sets of triangular facets acting as impenetrable obstacles. The simulated volume contains distributed spherical particles representing

zirconium dispersoids, and ellipsoidal particles representing manganese dispersoids, as shown in Figure 3.

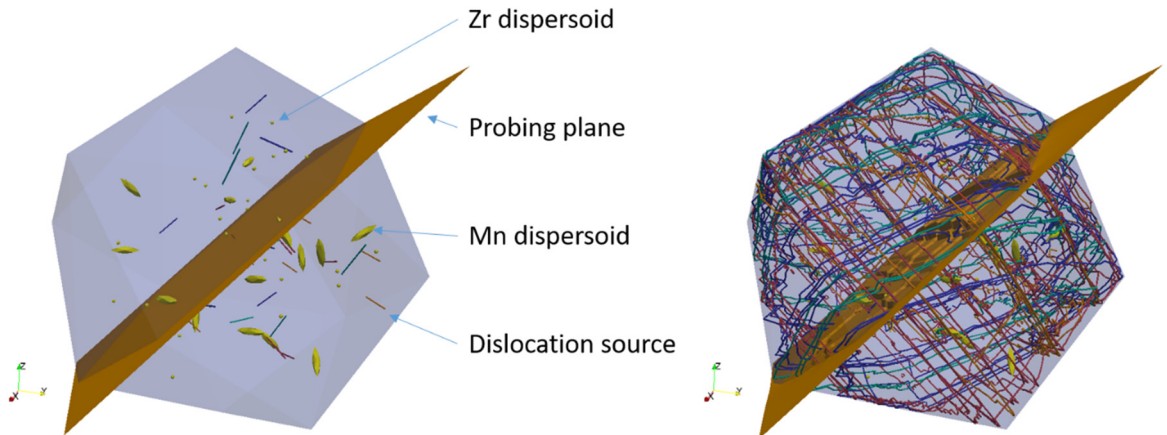

**Figure 3.** Initial and final configuration of a typical DDD simulation of a spherical volume of 5 μm diameter containing an assortment of small spherical zirconium dispersoids, elongated manganese dispersoids, and Frank Read dislocation sources. Strain localization will be estimated from the plastic steps printed in the probing plane inserted in the middle of the simulated volume.

The load is applied as a homogeneous tensile stress tensor applied along the $[-101]$ direction. The loading sequence is defined as a plastic strain rate monitoring mode at a rate $\dot{\varepsilon}_p = 80\,\mathrm{s}^{-1}$ along the tensile direction. Such a strain rate is low enough to allow the dislocations to reach an equilibrium position during the entire loading stage.

For each particle type, the dimension and volume fraction are varied. Configurations containing one type, both types, or no particles are systematically studied to understand the effect of each particle individually and together. To determine the effect of each particle, simulations are always carried out with the exact same configurations, i.e., with the same positions of the dispersoids and dislocation sources. Simulations with the same parameters but different configurations are used to determine the reproducibility of the results. The ranges of the investigated parameter values are presented in Table 3, together with the associated figures where the simulation results are presented.

**Table 3.** Range of the particle radii and volume fractions investigated in the DDD simulation campaign. The last column gives the figure where the results will be presented.

| Dispersoid | Volumic Fraction | Size (μm) | Results |
|:---:|:---:|:---:|:---:|
| Zr | 0.00003 | 0.02 | Figure 6 |
| Mn | 0.00045 | $0.5 \times 0.2$ | |
| Zr | 0 | 0 | |
| Zr | 0.00003 | 0.025 | Figure 7 |
| Zr | 0.00003 | 0.035 | |
| Zr | 0 | 0 | |
| Zr | 0.00002 | 0.025 | |
| Zr | 0.00003 | 0.025 | Figure 8 |
| Zr | 0.00004 | 0.025 | |
| Mn | 0 | 0 | |
| Mn | 0.00045 | $0.5 \times 0.2$ | Figure 9 |
| Mn | 0.00045 | $0.075 \times 0.3$ | |
| Mn | 0 | 0 | |
| Mn | 0.0003 | $0.5 \times 0.2$ | |
| Mn | 0.00045 | $0.5 \times 0.2$ | Figure 10 |
| Mn | 0.0006 | $0.5 \times 0.2$ | |

Particles are placed randomly in the volume, while avoiding overlapping each other. As for the manganese dispersoids, their orientation is randomly selected. While the random orientation does not represent what is usually found in aluminum alloys, it allows the results to be independent of the particle orientation.

Each simulation is run for about 15,000 steps to ensure that the plastic strain cumulated along the $[-101]$ loading direction is larger than $10^{-4}$. Analyses are all performed for a cumulated tensile strain of 0.01%. Since the amount of cumulated plastic strain simulated by DDD is much smaller than in the experiments (a few percent), the absolute values of the local shear cannot be compared. However, we think that the slip localizations found by DDD are the premises of the slip bands that will further develop for higher strains. Thus, DDD can help in the understanding of the strain localization mechanism.

The applied stress direction is $[-101]$, which simultaneously activates the dislocations from the slip systems C1, C5, A2 and A6 (See Table 4 for the slip system nomenclature). Those four slip systems are gliding on two different slip planes (A and C), which is evidenced at the probing plane by the two families of slip traces (see Figure 4).

**Table 4.** Slip system nomenclature following Schmid and Boas notation [26]. The letter indicates the slip plane normal, and the number corresponds to the Burgers vector direction.

| Vector | C1 | C5 | A2 | A6 |
|---|---|---|---|---|
| normal | $(-1-11)$ | $(11-1)$ | $(-111)$ | $(-111)$ |
| Burgers | $[011]$ | $[1-10]$ | $[0-11]$ | $[110]$ |

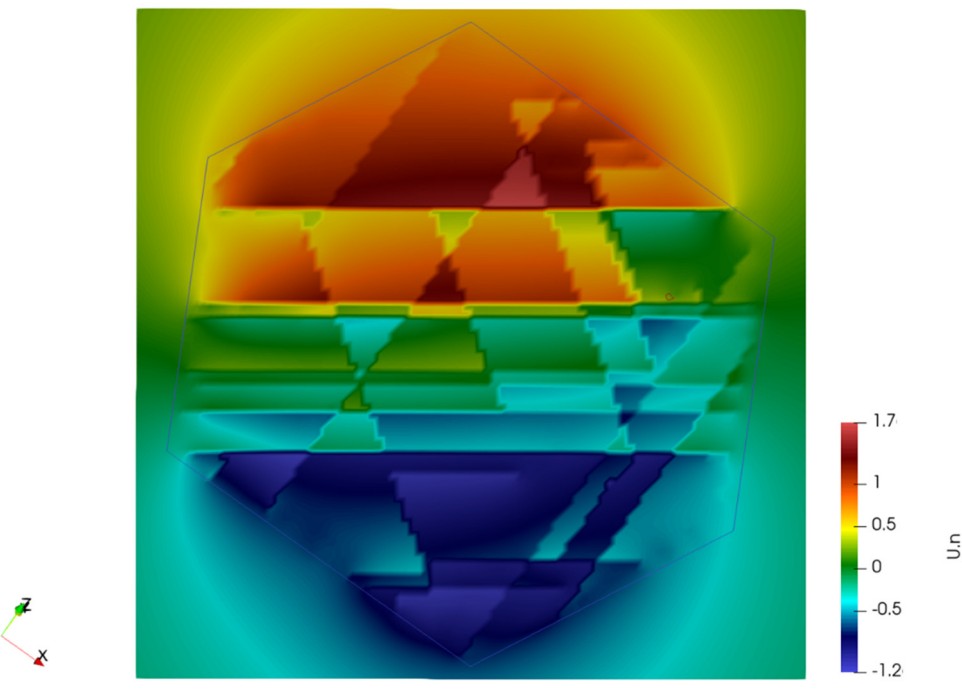

**Figure 4.** Typical plastic steps printed on the 5 μm × 5 μm probing plane defined in Figure 3 after $10^{-4}$ cumulated plastic tensile strain obtained by a DDD simulation. Scale is in nanometers. Colors correspond to the projection of the dislocation displacement field along the plane normal $[0-11]$.

Initially, all simulations contain the same number and type of dislocations, but those are placed in the volume randomly according to the particle position to avoid overlap. There are 24 dislocations initially introduced in the volume, six dislocations per each of the four active slip systems C1, C5, A6, and A2. These dislocations are 260 nm long, and their initial direction and type are randomly selected. A typical DDD simulation lasts 2.5 h on a

single Intel Xeon E5405 processor with eight threads. Once the simulations are completed, a 5 μm squared (0–11) calculation plane is inserted in the volume (see Figure 3) to visualize the spreading of the plastic strain inside the grain. The calculation plane is discretized into a grid of $100 \times 100$ points.

As shown in Figure 4, the plastic steps printed by the dislocations crossing the probing plane are then calculated using the optimized formula of the dislocation's displacement field from Fivel and Déprés [27]. Finally, five line profiles crossing the slip traces are extracted from the surface relief to quantify the magnitude of the strain localization. The local shear is then estimated at each grid point as the derivative of the profile height with the coordinate along with the line profile.

## 3. Results and Discussion

### 3.1. Experimental Results

Interferometric mapping results were post-treated, and ten surface profiles were extracted for each studied grain. For each profile, the mean height of the profile elements, $Pc$, is calculated. $Pc$ represents the height of the slip band extrusions. The higher $P_c$ is, the more the strain is concentrated on said bands. For each sample, the measurements were performed on a Brass oriented grain to avoid introducing an additional effect, due to the activation of different slip systems.

Results are pictured in Figure 5. For both alloys, samples containing both Zr and Mn dispersoids have lower $P_c$ values, showing that the simultaneous presence of both dispersoids tends to homogenize the plastic strain. With the addition of Mn dispersoids, the extrusion height is reduced by 14.7% for Li-Zr alloys and 18.8% for Li-Zr alloys showing that Mn is beneficial to reduce slip activity in slip bands.

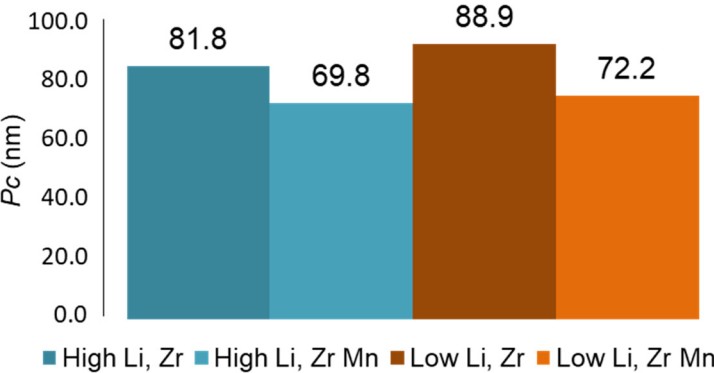

**Figure 5.** Average slip band extrusion height on slip bands measured for two aluminum alloys with two configurations of dispersoids on brass grains using interferometry.

Unexpectedly, it is also observed that the high lithium alloys have a slightly lower strain localization magnitude than the low lithium alloys (8% reduction for Li-Zr and 3.7% for Li-Zr-Mn). As discussed before, δ' (Al$_3$Li) precipitates are supposed to promote strain localization so that the measured effect might come from either a precision error, or from the fact that we study a single specific orientation, for which T$_1$ (Al$_2$CuLi) precipitation might be significantly sheared. Other material parameters, including the effect of Cu and Li content in solid solution, are not taken into account in this study and could also impact strain localization. It can be mentioned that other grain orientations can lead to the activation of different slip systems, and thus, different slip patterns.

Table 5 gives the measurements of the local shear (defined as *Pc/Psm*) in the case of High Lithium alloys. Here, again, it is found that the presence of Mn dispersoids reduces the strain localization by 28%.

**Table 5.** Local shear is estimated from the experimental measurements.

| Measured Quantities | No Mn | With Mn |
|---|---|---|
| *Psm* (µm) | 5.3 | 6.3 |
| **Standard Deviation** | **1.6** | **3.3** |
| *Pc* (µm) | 81.8 | 69.8 |
| **Standard Deviation** | **18.6** | **26.3** |
| Local shear = *Pc/Psm* | 15.4 | 11.1 |

### 3.2. Simulation Results

As for the simulations, three simulations are first conducted to verify if the experimental observations could be reproduced by DDD. A first simulation containing both Zr and Mn dispersoids is performed. Then a second simulation is run with only the Zr dispersoids, and finally, a third one is run with only Mn dispersoids. Size and volume fractions are given in Table 3.

Five line profiles taken along direction [0−11] are then extracted from the probing plane. The profile direction is chosen so that it is parallel to the slip traces of systems C1 and C5 so that it will give an estimate of the dislocation activity on systems A2 and A6, similarly to the experimental procedure where the line profile was perpendicular to a given slip band orientation.

The histograms of the local shear are plotted in Figure 6 together with the cumulative percentages, which better indicate the average values. The simulation with only Zr dispersoids admits an average local shear of $\gamma_{loc}$ = 0.001. The local shear measured from the simulation with only Mn dispersoids is decreased by 30%: 0.0007, and when both particles are present, the local shear reduces to $\gamma_{loc}$ = 0.0002 which is 5 times less than the case of solely Zr dispersoids. It can be concluded that the combined presence of Mn and Zr dispersoids is the most efficient manner to reduce the strain localization, which is consistent with the experiments presented in the previous section.

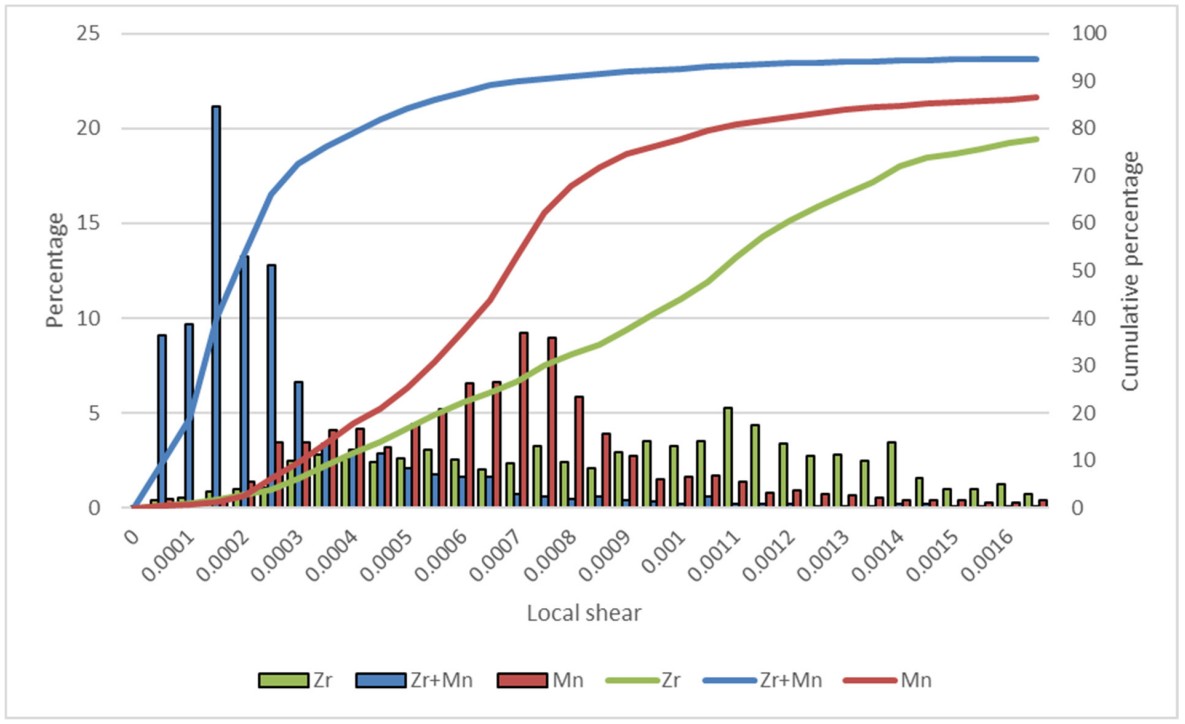

**Figure 6.** Local shear distribution from DDD simulations at $\varepsilon$ = 0.01% for three comparable simulations to identify the effect of each dispersoid type. For each of the three simulations, the position of the dispersoids and Frank–Read sources are kept the same.

In order to identify the effect of the volume fractions and the size of each dispersoid type, a DDD simulation campaign is now conducted. To access more data, the post-treatment of the surface relief is slightly modified: The direction of the line profiles is chosen so that it can intercept both slip traces. Figure 7 shows the effect of the size of the Zr dispersoids for two values of the sphere particle radius: 25 nm and 35 nm and for the given volume fraction fv = 0.00003. The case of a simulated volume without any dispersoids is also plotted for reference (labeled as 0 In Figure 7). It is shown that the smaller particle size (R = 25 nm) has nearly no effect on the local shear distribution. On the other hand, the bigger particles induce a decrease of the local shear by a factor 2. It can be concluded that there is a minimum particle size that can reduce strain localization. This could be explained by the role of the cross slip, which can be triggered on the Orowan dislocation loops deposited around the particles. When the particles are bigger, the curvature of the dislocation loops is bigger, and the cross-slip probability is increased. The dislocations emitted from the Frank–Read sources can then populate new slip planes that homogenize the plastic strain.

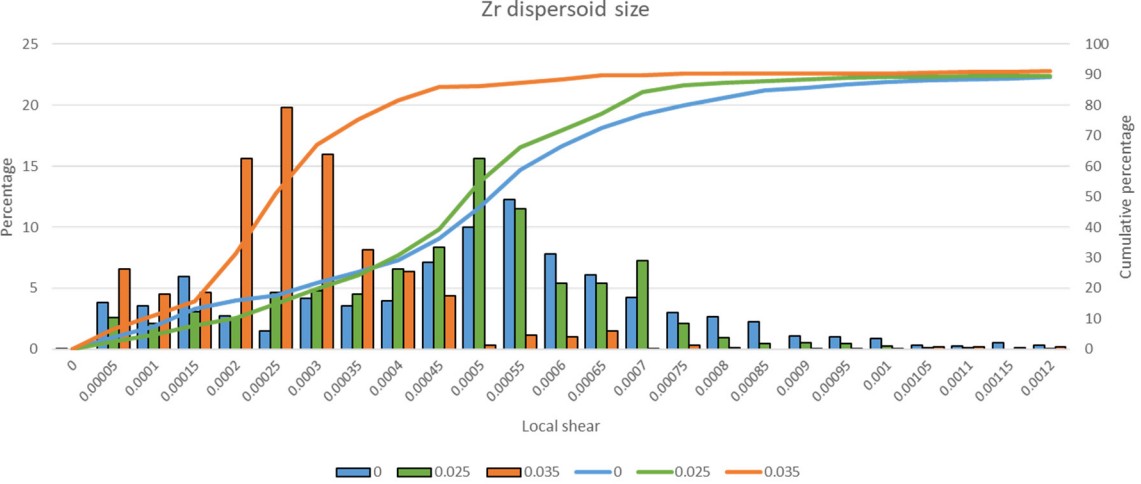

**Figure 7.** Local shear distribution from DDD simulations showing the isolated effect of Zr dispersoid size (fv = 0.003%). Radii are in μm. Label 0 refers to a simulation without any particle.

In Figure 8, the size of the Zr dispersoid is fixed to 25 nm, and the volume fraction is now increased to verify if there is a possibility to modify the local shear when the number of small Zr particles becomes high enough. It can be seen that for this small size of the Zr particle, changing the volume fraction from 0.002% to 0.004% has no effect on the slip dispersion. It is concluded again that the particle radius is too small to have an impact on the dislocation activity.

The same investigations are now conducted in the case of the Mn dispersoids, without the presence of Zr. Figure 9 shows the effect of the particle sizes on the local shear distribution for a fixed volume fraction fv = 0.045%. It is shown that increasing the particle size from 200 nm × 50 nm to 300 nm × 75 nm does not affect the strain localization. Looking at the simulation details, we could observe that the Orowan loops deposited around the particles are difficult to unpin because of the large shape factor of the Mn particles. When cross slip happens, the cross-slip segments are immediately in contact with the dispersoid and cannot escape. This locking phenomenon is enhanced when the size of the dispersoids is increased. In other words, the Mn dispersoids play the role of forest obstacles for the dislocation motion so that the strain is spread in the volume.

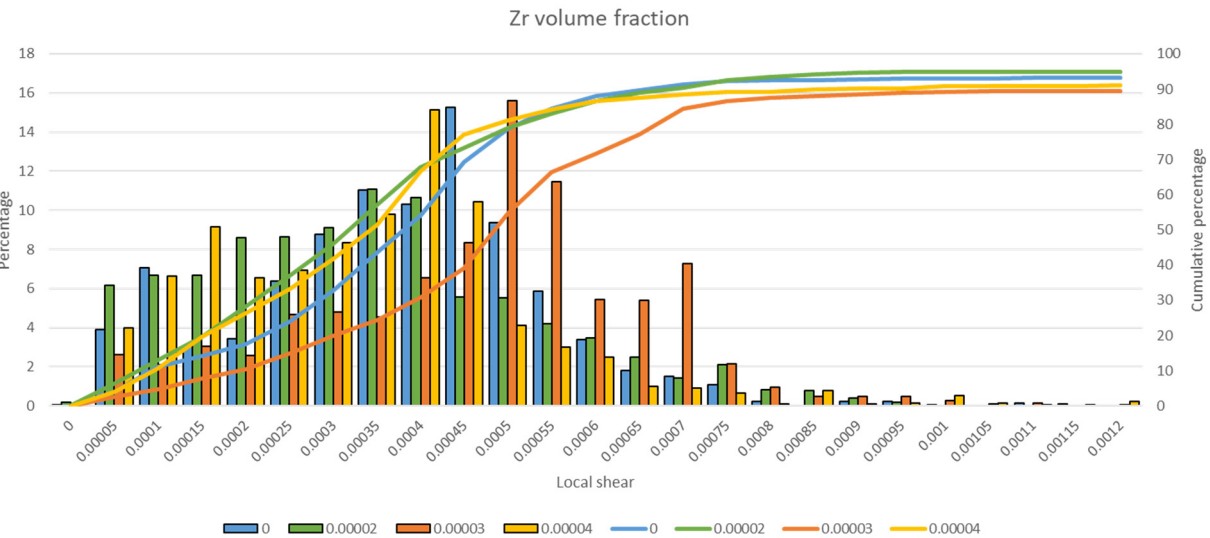

**Figure 8.** Local shear distributions from DDD simulations, showing the effect of Zr dispersoid volume fraction for particle size R = 25 nm. Label 0 refers to a simulation without any particle.

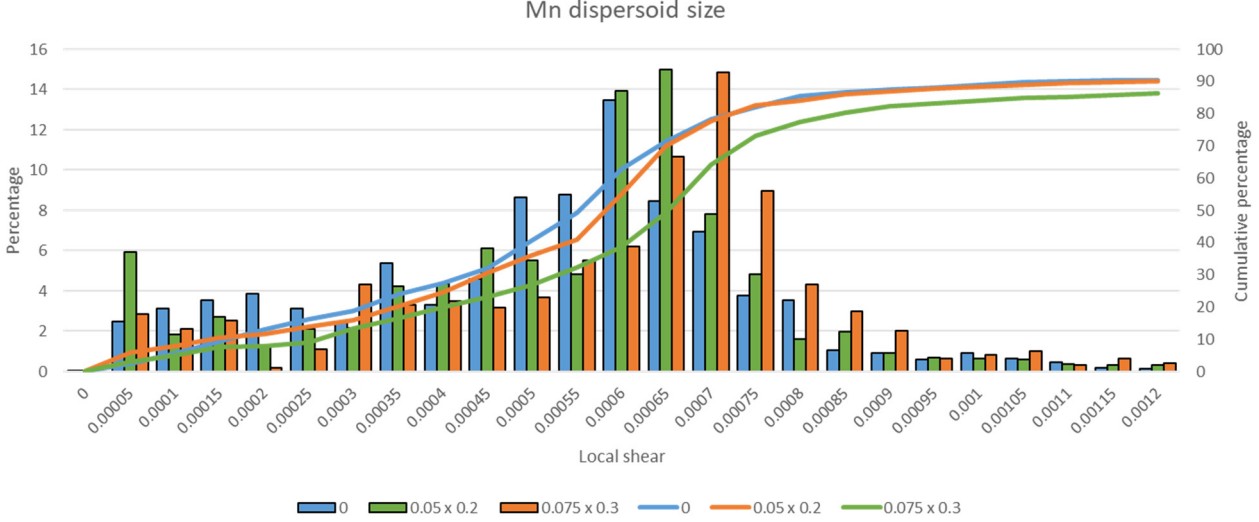

**Figure 9.** DDD simulation results from Mn dispersoid size on the local for fv = 0.0045%. Sizes are in μm. Label 0 refers to a simulation without any particle.

Finally, the volume fraction of Mn dispersoids is increased with a particle size fixed to 200 nm × 50 nm. Results are given in Figure 10 where it is shown that all the investigated volume fractions have a similar positive effect on the strain localization. One possible explanation is that when particles are more numerous, the decrease of strain localization induced by cross-slip and by-passing the particles is counterbalanced by the fact that the possible paths for the dislocations to move between the particles is limited in number, thus leading to strain localization within these paths.

Although the geometry of the Zr and Mn dispersoids is largely different, their impact on slip band behavior is similar. As mentioned beforehand, both dispersoids have a similar effect on the dislocation activity: The bigger they are, the more they can spread the plastic strain. The elongated shape of the Mn dispersoids is locking the by-passing mechanism by cross-slip. For zirconium spherical dispersoids, they can easily be looped by dislocations and promote cross-slip for the bigger sizes, which also leads to strain dispersion.

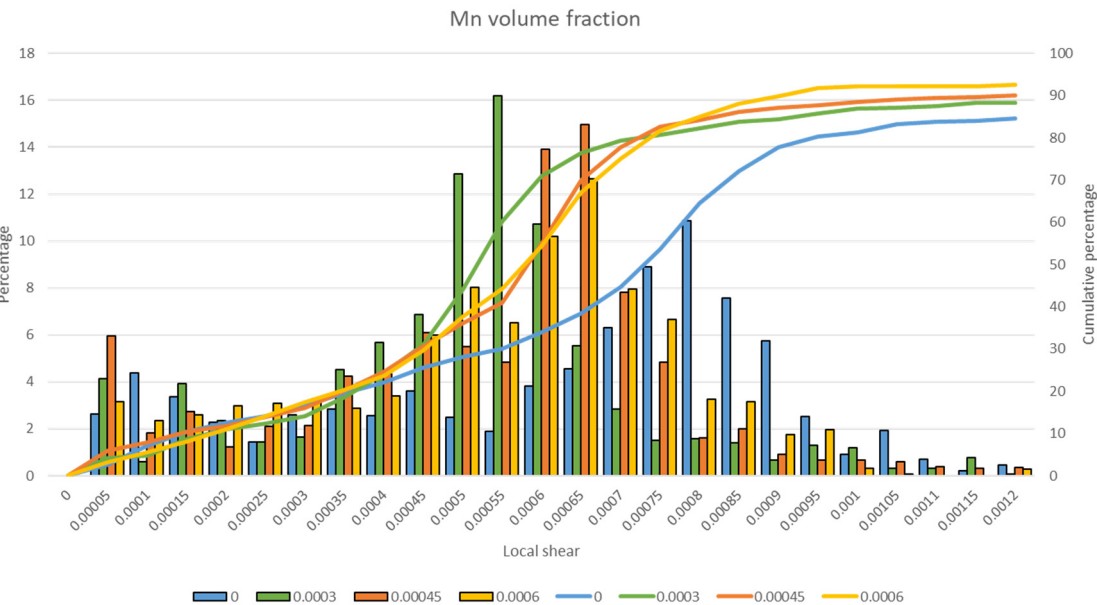

**Figure 10.** DDD simulation results from the volume fraction of Mn dispersoids on the local shear for a size 200 nm $\times$ 50 nm for the ellipsoid particles. Label 0 refers to a simulation without any particle.

## 4. Conclusions

In aluminum alloys, dispersoids affect the strain localization; the combined presence of Mn and Zr dispersoids in Al-Cu-Li alloys reduces the plastic strain amplitude in the slip bands by 28%. Discrete dislocations dynamics simulations show the same trend as in the experiments. It is also found that there exists a critical size for the Zr dispersoids that can lead to a decrease in the strain localization. Because of their aspect ratio, Mn dispersoids always tend to homogenize the plastic strain, thus reducing the strain localization. The best results should be achieved when both zirconium and manganese dispersoids are present in the alloy. The effect of the Mn volume fraction is negligible, but large Zr particles (R > 35 nm) are recommended.

**Author Contributions:** Conceptualization, E.J.C., M.F., J.C. and B.D.; methodology, J.C. and B.D.; software, E.J.C. and M.F.; validation, J.C.; formal analysis, E.J.C., M.F., J.C. and B.D.; investigation, E.J.C.; resources, J.C. and B.D.; data curation, E.J.C.; writing—original draft preparation, E.J.C.; writing—review and editing, M.F.; visualization, E.J.C.; supervision, M.F.; project administration, M.F., J.C. and B.D.; funding acquisition, J.C. and B.D. All authors have read and agreed to the published version of the manuscript.

**Funding:** This research received no external funding.

**Institutional Review Board Statement:** Not applicable.

**Informed Consent Statement:** Informed consent was obtained from all subjects involved in the study.

**Data Availability Statement:** The data presented in this study are available on request from the corresponding author. The data are not publicly available due to confidential issue from the industrial partner.

**Conflicts of Interest:** The authors declare no conflict of interest.

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
