# Peer review of "Effects of Manganese and Zirconium Dispersoids on Strain Localization in Aluminum Alloys"

_metals, doi:10.3390/met11020200_

Round 1

Reviewer 2 Report

This aim of this paper is exactly defined by the title of the paper. Two ways are followed for that purpose: experimental study and numerical one with dislocation dynamics simulation.

The approach is clear with parametric study depending on the type of dispersoid. Experimental results and simulations results give the same trends.

The paper is interesting and well presented. But, the originality of the paper should be better highlighted in the introduction: what is known and what is new? What do the simulations bring to the results? It should be recalled in the conclusion.

However, this paper is interesting and the reviewer recommends its publication with minor revisions.

Here are some specific comments:

  • The simulation and the experiment are not carried out under the same conditions (strain level, strain rate): this point should be discussed event it is well known that DDD are long time simulations.
  • 8 : There are errors in local strain values: 0.0007 is not 30% lower than 0.0001.
  • The results of the figure 6 is not discussed: is there an interaction between both particles that improves the strain homogenization?
  • The effect of grain size could be discussed too.
  • What is the initial roughness of the specimens? Even the authors said that the surface was prepared in order to avoid any interaction with localization or measurements, this point should be verified.
  • Minor comment: The wording or typing errors could be revised or verified: abstract last line “is the experimental”, “in the experimental”, measurement in place of measure, section 3.2 , “As for simulation …” replaced by “As for experiments …”, p.5 the sentence “As for the ellipsoid …” is not clear, P.5 “has been evolved enough”: enough for what ? blank spaces before unities. Punctuation in authors contribution should be revised, in the references also.

Reviewer 3 Report

The main topic of this paper is to investigate the size and shape effects of Mn and Zr dispersoids on strain localization in aluminum alloys. The results have some significance to metallurgical technology of aluminum alloys. However, there are several issues the authors may consider to address in order to improve the manuscript.
1. The last sentence of abstract is grammatically incorrect, “Discrete Dislocations Dynamics” appears only once in the abstract, and therefore, its abbreviation is not necessary.
2. Al-Cu-Li alloys with different Cu and Li contents are used to illustrate the effect of Mn and Zr dispersoids. The effects of Cu and Li contents on strain localization also should be clarified.
3. The work tends to evaluate the strain localization by measuring surface roughness induced by dislocation slip. However, this surface roughness is not only induced by strain localization but also by different activated slip systems in different grains. Rigorousness of this approach should be justified.
4. The important part of this paper is to investigate the size and shape effects of Mn and Zr dispersoids. The author should present the morphology of Mn and Zr dispersoids in the experimental results.
5. The experiment and the simulation results are not closely related.
6. The simulation results show a great difference in effecting strain localization between Mn and Zr dispersions. Besides size and shape, what are other inherent differences between these two hard brittle particle?

Round 2

Reviewer 1 Report

.

Reviewer 3 Report

No comments